

# Autumn ichthyoplankton assemblage in the Yangtze Estuary shaped by environmental factors

Hui Zhang, Weiwei Xian and Shude Liu

Key Laboratory of Marine Ecology and Environmental Sciences, Institute of Oceanology, Chinese Academy of Sciences, Qingdao, China
Laboratory for Marine Ecology and Environmental Science, Qingdao National Laboratory for Marine Science and Technology, Qingdao, China

## ABSTRACT

This study investigated the response of the ichthyoplankton community to environmental changes in the Yangtze Estuary using canonical correspondence analysis. Ichthyoplankton community and environmental data were recorded during the autumns of 1998, 2000, 2002, 2003, 2004, 2007 and 2009. Among the ichthyoplankton, the dominant larval and juvenile families were the Engraulidae, Gobiidae and Salangidae, and the most common eggs were from *Trichiurus lepturus*. The ichthyoplankton was identified via canonical correspondence analysis to three assemblages: an estuary assemblage dominated by *Chaeturichthys stigmatias*, a coastal assemblage dominated by *Engraulis japonicus* and *Stolephorus commersonii*, and an offshore assemblage dominated by *Trichiurus lepturus*. Regarding environmental factors in the Yangtze Estuary, suspended matter and surface seawater salinity were the main factors influencing the distributions of the different assemblages, while sediment from the Yangtze River during the rainy season and chlorophyll *a* were the principle drivers of the annual variances in the distribution of ichthyoplankton assemblages. Our aims in this study were to provide detailed characterizations of the ichthyoplankton assemblage in the autumns of seven years, examine the long-term dynamics of autumn ichthyoplankton assemblages, and evaluate the influence of environmental factors on the spatial distribution and inter-annual variations of ichthyoplankton assemblages associated with the Yangtze Estuary.

Corresponding authors
Hui Zhang, zhanghuifirst@gmail.com
Weiwei Xian, wwxian@qdio.ac.cn

## INTRODUCTION

The Yangtze Estuary, an area of complex and varied environmental conditions, is the link between the Yangtze River and the East China Sea (*Luo & Shen, 1994*). Sea currents in this area are impacted by freshwater input from the Yangtze River, the Huanghai Cold Water Column and the Taiwan Strait Current. Large volumes of nutrients are carried by the river; therefore, the Yangtze Estuary has high primary production and rich food resources (*Luo & Shen, 1994*). Various fish species feed, fatten and reproduce in this area. In recent years, with economic development and increasing population, resource and environmental exploitation has increased, resulting in enormous pressure on the ecology of the Yangtze

Estuary and adjacent waters. Such environmental changes will inevitably affect the survival of some species and, hence, the structure of biological communities in the Yangtze Estuary.

Ichthyoplankton, including fish eggs, larvae and juveniles, is the basis for the sustainable utilization of fishery resources (*Miller & Kendall, 2009*; *Zhang, Xian & Liu, 2015*), with survival directly influencing recruitment. Because of absent or weak independent swimming capabilities and a drifting nature, ichthyoplankton are poorly adapted to changing environmental conditions. Thus, ichthyoplankton are sensitive and susceptible to environmental changes and, therefore, a suitable indicator of variations in the ecological environment (*Boeing & Duffy-Anderson, 2008*). The ichthyoplankton assemblages in estuaries are complex both in species composition and distribution. Studies show that the organization of ichthyoplankton in estuarine systems is influenced by the interactive effects of a multitude of biotic and abiotic processes. Biological factors include the location, timing and manners of spawning, larval life history, larval behavior, rates of predation, and feeding (*Leis, 1991*; *Azeiteiro et al., 2006*). Physical factors include salinity (*Whitfield, 1999*), temperature (*Blaxter, 1992*), turbidity (*Islam, Hibino & Tanaka, 2006*), dissolved oxygen (*Rakocinski, LyczkowskiShultz & Richardson, 1996*), depth (*Wantiez, Hamerlin-Vivien & Kulbicki, 1996*), river flow (*Faria, Morais & Chicharo, 2006*), sediment characteristics and hydrographic events such as currents, winds, eddies, upwelling and stratification of the water column (*Gray, 1993*). A study of the assemblage structure of the ichthyoplankton community is necessary for the sustainable utilization of fishery resources, while knowledge of its dynamic characteristics can be utilized to monitor the health of an aquatic environment.

To date, investigations into ichthyoplankton communities in Yangtze Estuary have mainly concentrated on species composition, fauna distribution and seasonal variations (i.e., *Yang, Wu & Sun, 1990*), discussions on the relationship between the ichthyoplankton community and environmental factors (*Jiang, Shen & Chen, 2006*; *Liu, Xian & Liu, 2008*), or long term dynamics of spring ichthyoplankton assemblages (*Zhang, Xian & Liu, 2015*). Fish reproduce in the Yangtze Estuary throughout the year, and fishery resources are most abundant in autumn. However, there are few studies on the Yangtze Estuary as a breeding/nursery area for fish and the factors affecting survival/recruitment in the autumns of the seven years. This study investigated the impact of environmental factors on the ichthyoplankton community in the autumns of 1998, 2000, 2002, 2003, 2004, 2007 and 2009 in the Yangtze Estuary and the drivers of changes in the community using a community-ecology analytical method. The aim was to provide detailed characterizations of the ichthyoplankton assemblage in the autumns, reveal the seasonal difference between spring and autumn, and evaluate the influence of human activities on the recruitment of fishery resources.

## MATERIALS AND METHODS

### Study area and sample collection

Forty stations were established in the Yangtze Estuary (30°45′–32°00′N, 121°00′–123°20′E) during seven autumns (11/1998, 11/2000, 11/2002, 11/2003, 11/2004, 11/2007 and 11/2009), including five stations within the river and all others outside the river mouth (Fig. 1).

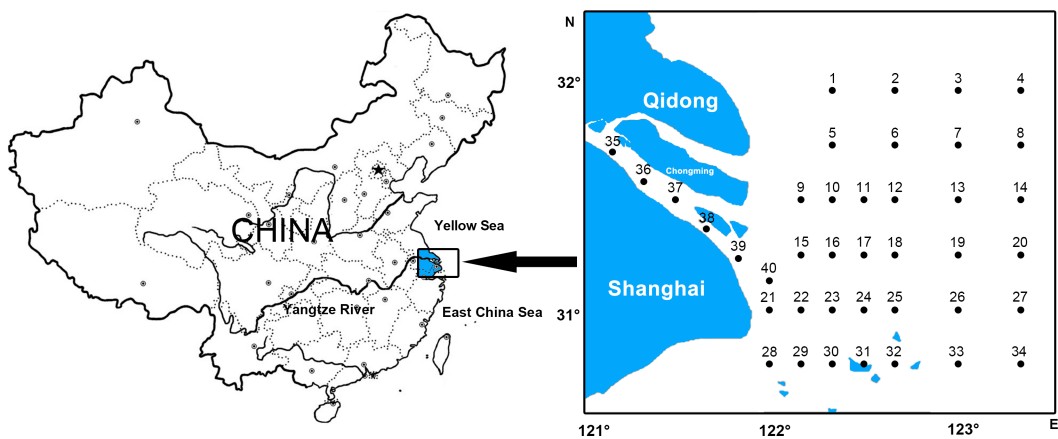

**Figure 1   Ichthyoplankton sampling stations in the Yangtze Estuary.**

Biological and oceanographic data were collected during seven fishery-evaluation cruises in seven autumns. Biological samples were collected by surface towing a larva net (0.8-m mouth diameter, 2.8-m length, 0.505-mm mesh at the body, and 0.505-mm mesh at the cod end) equipped with a flow meter. At each station, the net was towed at a depth of ~0.5 m from the surface for 10 min against the tidal flow, at a towing speed of approximately 2–3 knots. After completion of each tow, the nets were washed and the samples fixed and preserved in 5% buffered formaldehyde–seawater solution.

Geographical locations of sampling stations were provided by GPS, version Magellan 315. At each station, environmental variables, including depth (D), salinity (S), temperature (T), transparency (Trans), dissolved oxygen (DO), pH, total phosphorus (TP), total nitrogen (TN), total suspended matter (TSM), chemical oxygen demand (COD), and chlorophyll *a* (Chla), were determined according to the GB/T 12763-2007, Specification for Marine Monitoring (Supplemental Information 1). Because changes in freshwater flow into an estuary and adjacent areas affect nutrient levels, with consequences for primary productivity and associated trophic chains (*Morais, Chícharo & Chícharo, 2009*) and the above influences are always lagging behind. But as we could not get the data on freshwater flow directly, in the present study, we choose as a factor the total sediment (TSedi) entering the estuary from the Yangtze River during the rainy season brought by freshwater flow to do the analysis. The data on TSedi were provided by the Datong monitoring stations in the Changjiang Sediment Bulletin (http://www.cjh.com.cn/pages/nsgb.html).

## Data analysis

Ichthyoplankton were identified in the laboratory to the lowest possible taxonomic level. Numerical density for each species was standardized to catch per unit effort (CPUE) as abundance per 10 tow units.

The dominant species were determined using the Index of Relative Importance (IRI) developed by *Zhu, Liu & Sha (2002)*:

$$\text{IRI} = N^*100\%^*F^*100\%$$

$N*100\%$ and $F*100\%$ are the relative abundance and frequency of occurrence, respectively. The IRI of the dominant species should be greater than 100.

The ichthyoplankton assemblage structure and the relationship of the structure to environmental factors was analyzed via canonical correspondence analysis (CCA), which is utilized to visualize and describe the relationship between fish species and environmental variables (CANOCO Software, Version 4.5). Only species that occurred in > 1% of the catches, based on all species, were included in the analysis. Species abundance data were log10 $(x + 1)$ transformed to reduce the dominant effect of some species (*Clarke & Warwick, 2001*). A total of 33 species and 11 environmental factors were selected for CCA, using the following steps: (1) discard environmental variables with an inflation factor > 20 following each CCA iteration; (2) test marginal effects and unique effects of every environmental variable; (3) assess the contribution and significance of each variable using inter-set correlations between the CCA axes and environmental factors, as well as Monte Carlo permutation ($P < 0.05$) analysis simulation and the forward selection option within CANOCO (CANOCO Software, Version 4.5, *Ter Braak, 1987*).

## RESULTS

A total of 969 ichthyoplankton constituting 33 species from 19 families and 10 orders were collected during the seven autumn samplings in the Yangtze Estuary, including 226 fish eggs and 743 larvae and juveniles (Table 1). Perciformes was the most abundant fish species, while species from the Order Clupeiformes were the second most common. The dominant larval and juvenile species were from the Engraulidae, Gobiidae and Salangidae families. The most abundant eggs where those of the largehead hairtail, *Trichiurus lepturus* (Table 2). Raw data of the present work could be download as the supplements.

Results of species–station and environment–station matrix analysis showed that the sum of all canonical eigenvalues was 2.844 (Table 3). The cumulative percentage variance of species totaled 17.5% and the cumulative percentage variance of species–environment was 72.2%. Eigenvalues of the first four axes was 0.696 (CCA1), 0.549 (CCA2), 0.419 (CCA3) and 0.389 (CCA4); the eigenvalues of the first two axes were moderately high. The correlation between the first two axes and environmental factors was the highest with correlation coefficients of 0.884 and 0.788, respectively. The first two CCA axes explained 10.6% of the cumulative percentage variance in species and 43.8% of cumulative percentage variance in species–environment (Fig. 2).

Through Monte Carlo tests of $F$-ratios ($P < 0.05$), TSM, S, TSedi from the Yangtze River during the rainy season, Chla and PH of surface seawater were the most active environmental parameters affecting ichthyoplankton. Total variation due to TSM and TSedi was higher than other environmental factors. The patterns of D, TN, TP, TSM and S were highly correlated with the first CCA axis, and this CCA1 axis represented a spatial gradient from the Yangtze River to the sea. There was a high correlation between the second axis and TSedi, i.e., sediment from the Yangtze River during the rainy season, and Chla; the CCA2 axis represented a temporal gradient (Table 4).

TSM was negatively correlated with S, D and transparency (Trans), representing a decrease in TSM from the river to the Offshore with increasing salinity, depth and

Zhang et al. (2016), *PeerJ*, DOI 10.7717/peerj.1922

**Table 1** Ichthyoplankton species assemblage revealed by CCA in the Yangtze Estuary in the autumns.

| Species | Abbreviation | Species assemblage | 1998 | 2000 | 2002 | 2003 | 2004 | 2007 | 2009 | Percentage caught in station areas[a] | | |
|---|---|---|---|---|---|---|---|---|---|---|---|---|
| | | | | | | | | | | Estuary area | Coastal area | Offshore area |
| *Hypoatherina valenciennei* | Hval | — | | | | | | ✓ | ✓ | | | 0.65(2) |
| *Thryssa kammalensis* | Tkam | + | | | | | | ✓ | ✓ | | 1.11(3) | 3.23(5) |
| *Larimichthys crocea* | Lcro | + | | | ✓ | | | | | | 0.56(2) | 3.23(6) |
| *Trichiurus lepturus* | Tlep | + | | | ✓ | | | | ✓ | 0.39(1) | 1.67(3) | 58.71(18) |
| *Coilia nasus* | Cnas | × | | | | | | ✓ | ✓ | 0.39(1) | 0.84(3) | |
| *Sparidae* sp. | Spar | — | | | | | ✓ | | | | 0.28(1) | |
| *Pholis fangi* | Pfan | — | | | ✓ | | | | | | 0.28(1) | |
| *Coilia mystus* | Cmys | × | ✓ | | | | | ✓ | | 0.79(1) | 2.79(4) | 0.32(1) |
| *Pseudolaubuca engraulis* | Peng | ▲ | | ✓ | | | | | | 2.36(3) | | |
| *Odontamblyopus rubicundus* | Orub | — | | | ✓ | | | | | | 0.28(1) | |
| *Minous monodactylus* | Mmon | — | | | | | | ✓ | | | | 0.65(1) |
| *Saurida undosquamis* | Sund | — | | | | | ✓ | | | | 0.28(1) | |
| *Setipinna taty* | Stat | — | | | | | | ✓ | | | 0.28(1) | |
| *Syngnathus acus* | Sacu | × | | ✓ | | | | ✓ | ✓ | | 3.62(6) | |
| *Stolephorus commersonnii* | Scom | × | ✓ | ✓ | | ✓ | ✓ | | | | 23.40(9) | 4.84(8) |
| *Harpadon nehereus* | Hneh | + | | ✓ | ✓ | ✓ | ✓ | ✓ | ✓ | | 1.67(5) | 3.87(7) |
| *Lateolabrax japonicus* | Ljap | + | | ✓ | ✓ | | | | | | | 2.90(5) |
| *Pseudorasbora parva* | Ppar | ▲ | ✓ | | | | | | ✓ | 11.42(6) | | |
| *Chaeturichthys stigmatias* | Csti | ▲ | | | | | | ✓ | ✓ | 57.48(3) | 2.51(2) | |
| *Omobranchus elegans* | Oele | — | | | | | | | ✓ | | | |
| *Benthosema pterotum* | Bpte | + | | | ✓ | | | ✓ | | | 0.28(1) | 3.23(3) |
| *Channa asiatica* | Casi | — | | ✓ | ✓ | | | | | | 0.28(1) | 1.29(4) |
| *Salanx prognathus* | Spro | ▲ | | | | | | ✓ | | 22.05(1) | 29.81(6) | |
| *Stolephorus zollingeri* | Szol | + | | | | ✓ | | | | | | 0.65(2) |
| *Acentrogobius caninus* | Acan | — | | | ✓ | | | | | 0.39(1) | | |
| *Argyrosomus japonicus* | Ajap | — | | | ✓ | | | | | | | 0.32(1) |

Zhang et al. (2016), *PeerJ*, DOI 10.7717/peerj.1922

**Table 1** (*continued*)

| Species | Abbreviation | Species assemblage | 1998 | 2000 | 2002 | 2003 | 2004 | 2007 | 2009 | Percentage caught in station areas[a] | | |
|---|---|---|---|---|---|---|---|---|---|---|---|---|
| | | | | | | | | | | Estuary area | Coastal area | Offshore area |
| *Engraulis japonicus* | *Ejap* | × | | √ | √ | √ | √ | √ | | | 20.06(8) | 10.65(10) |
| *Apogon lineatus* | *Alin* | — | | | | | √ | | | | | 0.65(1) |
| *Pseudolaubuca sinensis* | *Psin* | — | | √ | | | √ | √ | | 2.36(3) | | |
| *Salanx ariskensis* | *Sari* | × | | √ | √ | | √ | | | 2.36(1) | 8.64(7) | 1.29(4) |
| *Hyporhamphus limbatus* | *Hlim* | — | | | | | | | √ | | | 0.97(1) |
| *Callionymidae* sp. | *Call* | — | | | √ | | | | | | | 0.32(1) |
| species1 | *Spe1* | × | | | √ | | | √ | | | | 2.26(2) |

**Notes.**

▲, Estuary assemblage; ×, Coastal assemblage; +, Offshore assemblage; —, Rare species determined by IRI , which were excluded for the species assemblage analysis; √, The species was collected in the year.

[a]Numbers in parentheses indicated the amounts of stations that the species was caught.
**Table 2   Dominant species determined by the IRI.**

| Species | IRI | | | | | | |
|---|---|---|---|---|---|---|---|
| | 1998 | 2000 | 2002 | 2003 | 2004 | 2007 | 2009 |
| *Trichiurus lepturus* | | | 807.10 | | | | 3145.50 |
| *Salanx prognathus* | | | | | | 710.31 | |
| *Stolephorus commersonnii* | 334.49 | 4.43 | | 1686.24 | 713.97 | | |
| *Engraulis japonicus* | | 177.38 | 8.87 | 6.02 | 4.43 | 584.08 | |
| *Chaeturichthys stigmatias* | | | | | | 329.48 | 1.05 |
| *Salanx ariskensis* | | 345.90 | 53.22 | | 319.29 | | |
| *Pseudorasbora parva* | 459.93 | | | | | | 22.08 |
| *Harpadon nehereus* | | 124.17 | 26.61 | 3.01 | 4.43 | 6.42 | 42.05 |
| *Larimichthys crocea* | | | 141.91 | | | | |

**Table 3   Results of canonical correspondence analysis relating ichthyoplankton abundance data to environmental variables in the Yangtze Estuary in the autumns.**

| | CCA axes | | | | Total inertia |
|---|---|---|---|---|---|
| | 1 | 2 | 3 | 4 | |
| Eigenvalues | 0.696 | 0.549 | 0.419 | 0.389 | 11.741 |
| Species-environment correlations | 0.884 | 0.788 | 0.680 | 0.689 | |
| Cumulative percentage variance | | | | | |
| of species data | 5.9 | 10.6 | 14.2 | 17.5 | |
| of species-environment relation | 24.5 | 43.8 | 58.5 | 72.2 | |
| Sum of all unconstrained eigenvalues | | | | | 11.741 |
| Sum of all canonical eigenvalues | | | | | 2.844 |

**Table 4   Conditional effects and correlations of environmental variables with the CCA axes.**

| Environmental factors | LambdaA | *P* value | *F* value | Axis1 | Axis2 |
|---|---|---|---|---|---|
| TSM | 0.49 | 0.002 | 4.8 | 0.53 | −0.44 |
| TSedi | 0.44 | 0.002 | 4.48 | 0.33 | 0.52 |
| S | 0.36 | 0.002 | 3.64 | −0.58 | 0.08 |
| Chla | 0.32 | 0.002 | 3.45 | 0.21 | 0.42 |
| PH | 0.31 | 0.01 | 3.27 | 0.05 | −0.06 |
| T | 0.21 | 0.056 | 2.28 | 0.16 | 0.11 |
| D | 0.17 | 0.058 | 1.94 | −0.61 | 0.19 |
| TP | 0.17 | 0.06 | 1.83 | 0.56 | −0.35 |
| Trans | 0.15 | 0.056 | 1.72 | −0.46 | 0.38 |
| TN | 0.12 | 0.186 | 1.38 | 0.54 | −0.17 |
| DO | 0.1 | 0.378 | 1.13 | 0.22 | −0.15 |
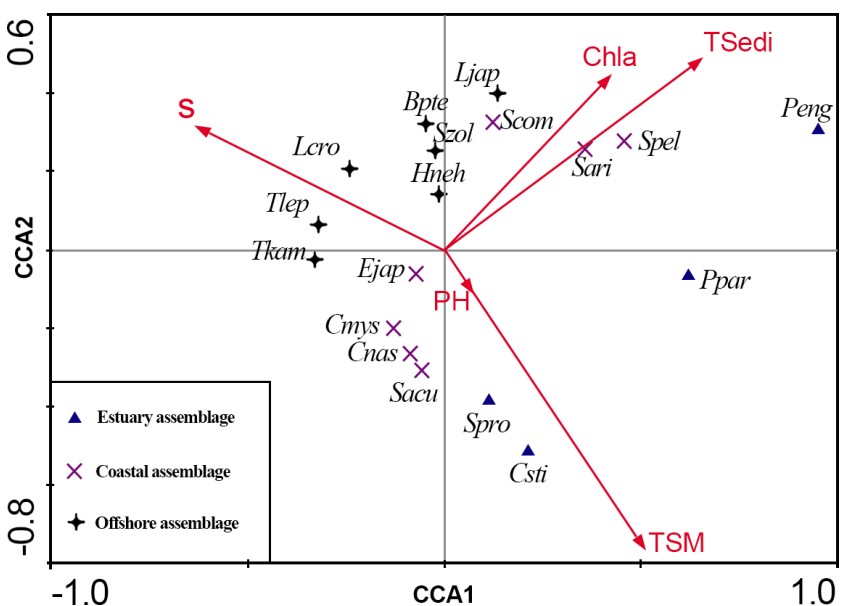

**Figure 2  CCA biplot of ichthyoplankton species in the Yangtze Estuary in the autumns.**

transparency. There was a positive correlation between TSedi and Chla, both impacting on changes in the annual ichthyoplankton assemblage. There were combined effects of environmental factors in the CCA biplot of ichthyoplankton species. The resulting hierarchical classification yielded both species and station groups.

Three species groups were revealed by CCA (Table 1 and Fig. 2):

Estuary assemblage: This group included four species (*Chaeturichthys stigmatias*, *Salanx prognathous*, *Pseudolaubuca engraulis* and *Pseudorasbora parva*), distributed in an area of low salinity and high suspended matter;

Coastal assemblage: This group included seven species (such as *Stolephorus commersonii*, *Engraulis japonicus*, *Salanx ariskensis*), distributed in brackish water;

Offshore assemblage: This group included seven species (such as *Trichiurus lepturus*, *Thryssa kammalensis*, *Harpadon nehereus*), distributed on top-left of the CCA biplot and correlated with salinity.

Three station areas were also revealed by CCA (Table 1, Figs. 3A and 3B):

I. Estuary area consisting of five sites: 35, 36, 37, 38 and 39, distributed in the river system. In this group, *Chaeturichthys stigmatias* (57.48%) was the most abundant species, followed by *Salanx prognathous* (22.05%). Salinity in this area was $2.34 \pm 1.1‰$, and the total suspended matter was $121.66 \pm 61.54$ mg/L.

II. Coastal area consisting of 16 sites: 9–11, 15–18, 21–24, 28–31 and 40, distributed in water near the mouth of the Yangtze Estuary. The dominant species in this group included *Salanx prognathous* (29.81%), *Stolephorus commersonii* (23.40%) and *Engraulis japonicus* (20.06%). Salinity in this area was $19.41 \pm 8.17‰$ and the total suspended matter were $134.77 \pm 240.10$ mg/L, the highest of three groups.

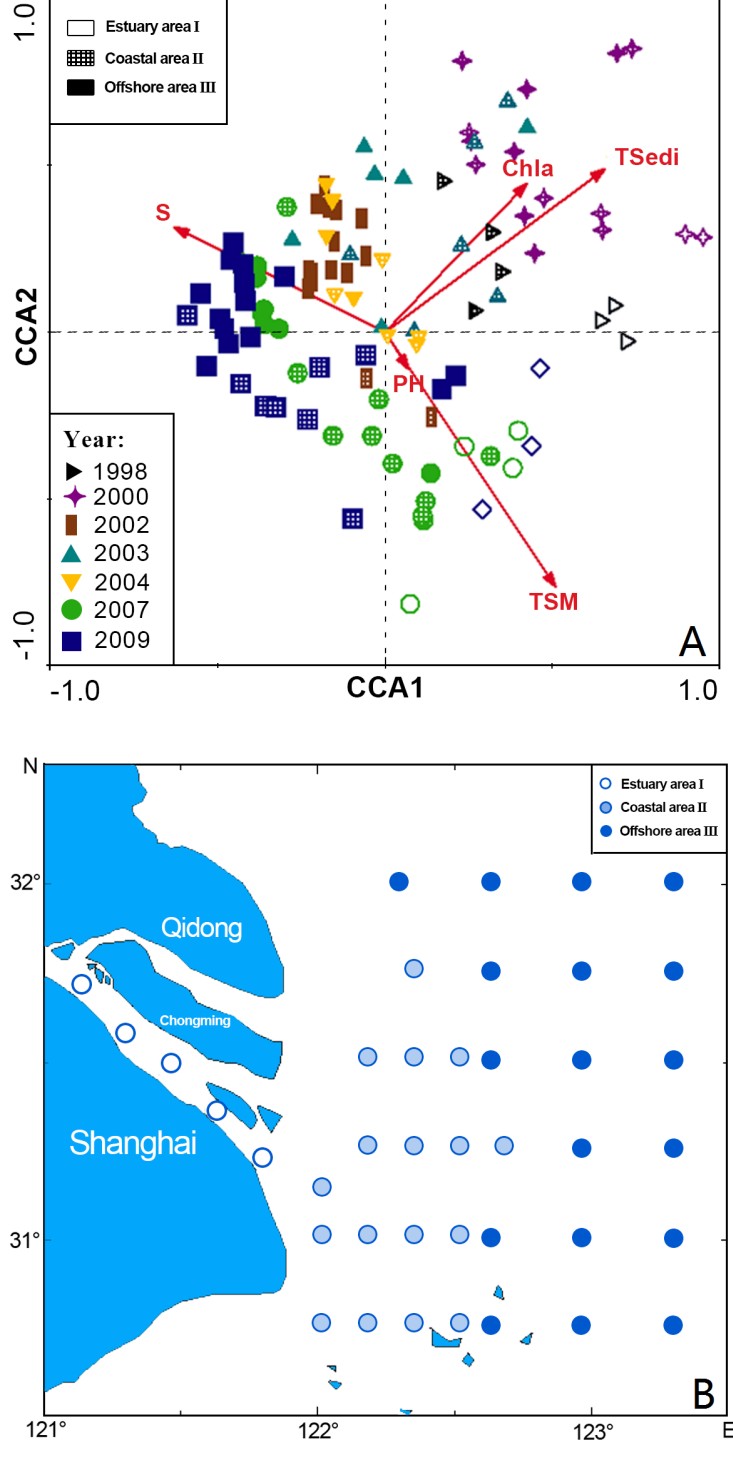

**Figure 3** CCA biplot of sampling stations for ichthyoplankton in the Yangtze Estuary in the autumns.

III. Offshore area consisting of 19 sites: 1–8, 12–14, 19, 20, 25–27 and 32–34, located in offshore areas outside the Yangtze Estuary. *Trichiurus lepturus* (58.90%) and *Engraulis japonicus* (10.68%) were the dominant species. Salinity in this area was $19.41 \pm 8.17\text{‰}$ (the highest of three groups), and the total suspended matter was $7.78 \pm 13.99$ mg/L (the lowest of three groups).

In all, environmental conditions were significantly different among the three stations groups, and the distributions of ichthyoplankton varied with environmental factors from the estuary to the offshore area.

## DISCUSSION

As a result of the interaction between biological and abiotic factors, in this study, coastal and offshore ichthyoplankton were dominant in the Yangtze Estuary during the November sampling periods. Compared with the characteristics of the ichthyoplankton assemblage within the Yangtze Estuary in springs (*Zhang, Xian & Liu, 2015*), species composition differed in autumn and the species within the three groups were also different. The *S. commersonii* assemblage group remained unchanged compared to the results in springs, although the seawater temperature within the estuary water may have changed between the springs and autumns (*Zhang, Xian & Liu, 2015*). Moreover, formation of the ichthyoplankton assemblage and maintenance of the community structure was impacted by the spawning patterns and distribution of fish species (*Duffy-Anderson et al., 2006*; *Rodriguez, 2008*), but the majority of fish species do not spawn in autumn.

Seasonal patterns of abundance of ichthyoplankton are mainly linked to reproductive strategies of adult population and phases of their life cycles, which, in turn are often associated with oceanographic and meteorological features (*Hernández-Miranda, Palma & Ojeda, 2003*). Most of the species collected in Yangtze Estuary spawned in spring or early in the summer, and both the species and the amount of ichthyoplankton were decreased in autumn. The present study and other studies supported the above point (i.e., *Yang, Wu & Sun, 1990*; *Jiang & Shen, 2006*; *Yu & Xian, 2009*). That is one of the reasons that there are significant differences in ichthyoplankton assemblages between spring and autumn.

The ability of larval and juvenile fish to survive the pelagic phase and migrate to a suitable adult habitat may be dependent on their ability to regulate their dispersal or migration (*Muhling & Beckley, 2007*); larvae of different species tend to co-occur as a result of parallels in their life histories (*Gray & Miskiewicz, 2000*). *Duffy-Anderson et al. (2006)* found that exposure to similar hydrographic processes and convergence with a similar resource can all lead to discrete groupings of co-occurring larval fish species. The ichthyoplankton assemblage was grouped in accordance with suspended matter and surface seawater salinity within the Yangtze Estuary in autumn, and the difference in distribution within each group was the result of sediment from the Yangtze River during the rainy season and chlorophyll *a*.

CCA analysis was utilized to reflect the correlations between community groups and environmental factors. The ichthyoplankton assemblage was divided into three groups: an estuary, coastal and Offshore assemblage. Compared with the ichthyoplankton assemblage pattern in spring after 2004 (*Zhang, Xian & Liu, 2015*), there were three groups in autumn

and the distribution of the coastal assemblage was significantly enlarged compared with the estuary or offshore assemblage. Suspended matter and surface seawater salinity were the most important environmental factors influencing the ichthyoplankton assemblage pattern. However, total sediment from the Yangtze River during the rainy season and chlorophyll *a* were the major factors affecting changes in the annual ichthyoplankton assemblage. Other environmental factors, such as temperature, nutrients, depth, dissolved oxygen and chemical oxygen demand, had no significant impact on ichthyoplankton.

In estuaries, there are significant gradients in salinity and turbidity (*Islam, Hibino & Tanaka, 2006*). The spatial distribution in salinity was impacted by runoff, flow rate in the river, tidal intensity and topographic conditions (*Zhou et al., 2007*). Salinity is the most important environmental condition for estuarine organisms, influencing not only growth, development and reproduction of ichthyoplankton but also the temporal and spatial distribution of larval fish assemblages (*Zhang, Xian & Liu, 2015*). Fish species, as well as fish eggs and larvae, differ in their ability to adapt to salinity. Offshore, low-salinity seawater contributed to reproduction and egg hatching in fish species, while the salinity and temperature range to which larvae and juveniles were adapted was wider than that for fish eggs (*Jiang, Shen & Chen, 2006*).

The salinity was negatively correlated with suspended matter (Fig. 3), which was in agreement with other research results from the Chikugo Estuary (*Islam, Hibino & Tanaka, 2006*). This is an important characteristic of estuarine ecosystems. Salinity gradually increases from the river to the sea while suspended matter gradually decreases, resulting in an ecological gradient for salinity and turbidity due to the inflow of fresh water carrying a large amount of sediment. In addition, suspended matter was positive correlated with total phosphorus and total nitrogen. Seawater areas with high suspended matter content had abundant nutrients and, therefore, large numbers of food organisms. Such areas can result in increased predation rates of larval fish (*Islam, Hibino & Tanaka, 2006*) but also provide shelter from predation (*Barletta-Bergan, 2002*).

Freshwater inflow is the critical factor determining abiotic and biotic variability (*Morais, Chícharo & Chícharo, 2009*), and has an important impact on the distribution and abundance of ichthyoplankton within an estuarine ecosystem (*Faria, Morais & Chicharo, 2006*). Changes in freshwater flow into an estuary and adjacent areas affect nutrient levels, with consequences for primary productivity and associated trophic chains (*Morais, Chícharo & Chícharo, 2009*). The CCA results demonstrate that total sediment entering the Yangtze Estuary during the rainy season was one of the most important factors influencing changes in the annual ichthyoplankton assemblage. River freshwater inflow and sediment levels entering the Yangtze Estuary were impacted by large-scale, water-conservancy construction projects in the upper reaches of the Yangtze River and rainfall, especially the construction of the Three Gorges Dam. Some of the water, and thus sediment entering the Yangtze Estuary was intercepted and the seasonal flow allocation of the Yangtze River was changed. In this study, no correlation was found between sediment from the Yangtze River entering the sea during the rainy season and suspended matter in the surface seawater, i.e., suspended matter in surface seawater is not related to river flow or sediment entering the Yangtze Estuary during the whole rainy season.

Chlorophyll *a* also influenced changes in the annual ichthyoplankton assemblage, and was positively correlated with sediment entering the Yangtze Estuary. Chlorophyll *a* is a relative accurate reflection of the standing crop of phytoplankton; the greater the phytoplankton biomass, the higher the primary productivity (*Whitfield, 1999*). There is a close relationship between chlorophyll *a* and nutrients from freshwater entering the estuary. When nutrient levels were high, chlorophyll *a* levels in the estuary were also high, and food resources, which are necessary for the development and growth for larvae and juveniles, were abundant. Rich food resources also increased survival and successful foraging rates. From the standpoint of trophodynamics, high chlorophyll *a* levels provided a food indicator for recruitment in the early life-stage of fish species (*Wan, Huang & Zhan, 2002*). Figure 2 demonstrates that chlorophyll *a* had a major impact on the distribution of three freshwater fish, *Stolephorus commersonii*, *Salanx ariskensis* and *Engraulis japonicus*. *Wan, Huang & Zhan (2002)* also reported a close relationship between chlorophyll *a* and the distribution of eggs, larvae and juveniles of *E. japonicus*. Moreover, a study on the distribution and variation in chlorophyll *a* demonstrated the relationship between phytoplankton and the environment and changes in environmental conditions (*Ma et al., 2004*).

PH is another important environmental factor influencing the distribution of an ichthyoplankton assemblage. *Vazzoler (1996)* recorded a positive correlation between PH and the density of some larval fish. Although it is unclear how PH affects reproductive processes, it can induce spawning in some fish species. *Baumgartner, Nakatani & Baumgartner (1997)* noted that the preference of some species for water with a weak acidic pH may be an adopted behavior. In addition to the above-mentioned environmental factors, other factors such as temperature, total phosphorus, total nitrogen and chemical oxygen demand can affect ichthyoplankton assemblages to some degree but not significantly so. However, additional factors such as ocean currents and zooplankton/phytoplankton abundance, which were not investigated in this study and may affect ichthyoplankton (*Ignaciovilchis, Ballance & Watson, 2009*), require further detailed study.

## CONCLUSIONS

There are three species groups and three station areas revealed by CCA based on the biological and environmental data in the seven autumns in the Yangtze Estuary which was consistent with the results of the springs and other estuarine systems.

SPM and S were the most important environmental factors influencing the ichthyoplankton assemblage pattern. However, TSedi from the Yangtze River during the rainy season and Chla were the major factors affecting changes in the annual ichthyoplankton assemblage. Other environmental factors, such as T, nutrients, D, DO and COD, had no significant impact on ichthyoplankton.

## ACKNOWLEDGEMENTS

We are grateful to the editors and two anonymous reviewers for their constructive feedback and concerning on our work. We also thank Mr. Yan Qin for editing the figures.

### Funding

This study was supported by National Natural Science Foundation of China (No. 41406136, No. 31272663, No. 41176138 and U1406403). The funders had no role in study design, data collection and analysis, decision to publish, or preparation of the manuscript.

### Grant Disclosures

The following grant information was disclosed by the authors:
National Natural Science Foundation of China: 41406136, 31272663, 41176138, U1406403.

### Competing Interests

The authors declare there are no competing interests.

### Author Contributions

- Hui Zhang conceived and designed the experiments, performed the experiments, analyzed the data, contributed reagents/materials/analysis tools, wrote the paper, prepared figures and/or tables, reviewed drafts of the paper.
- Weiwei Xian conceived and designed the experiments, performed the experiments, contributed reagents/materials/analysis tools, reviewed drafts of the paper.
- Shude Liu performed the experiments, contributed reagents/materials/analysis tools, reviewed drafts of the paper.

### Data Availability

The raw data has been supplied as Data S1 and S2

### Supplemental Information

Supplemental information for this article can be found online at http://dx.doi.org/10.7717/peerj.1922#supplemental-information.

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
