# Peer review of "Autumn ichthyoplankton assemblage in the Yangtze Estuary shaped by environmental factors"

_PeerJ, doi:10.7717/peerj.1922_

## Round 0.1 · original submission · Major Revisions

Please, revise the manuscript according to the suggestions from the two reviewers.

Reviewer 1 ·

Basic reporting

There are some mistypes, references found in text but not cited in references and ambiguous sentences, although figures and tables are clearly shown. The aim shown in the introduction is unclear.

Experimental design

Analyzing spatiotemporal distribution of ichthyoplankton in relation to environmental conditions is enough because the methods are followed those done in the previous study. However, description on how to measure and select environmental conditions is not adequate.

Validity of the findings

It might be concerned that amount of ichthyoplankton analyzed in this study (under 1000 during seven years) might be too small to detect temporal variability compared with the previous study in spring (over 8000 individuals during four years). Seasonal differences on ichthyoplankton abundances may need to be discussed in this manuscript.

Additional comments

As commented at the basic reporting, the aim shown in the manuscript is unclear. Ichthyoplankton assemblages in spring in the study area have been reported in Zhang et al. (2015), and this manuscript demonstrates those in autumn based on the same methods. Therefore, seasonal difference between spring and autumn should be emphasized as the aims of this study. The title of this manuscript also should be addressed ichthoplankton assemblages in autumn rather than emphasizing the survey period.
How to measure and select environmental conditions should be carefully described in materials and methods. Vertical distribution of abiotic and biotic environmental conditions is highly variable in estuarine waters. Authors may need to show in which stratum environmental variables were measured in the water column. Furthermore, no explanation is found how relevant sediment entering the estuary during the rainy season (early summer), which is different from the current survey season, to ichthyoplankton assemblage.
Differences in ichthyoplankton assemblages between spring and autumn are discussed in the current manuscript, however reasons for the differences are not clear. Why the coastal assemblages in autumn are enlarged compared to those in spring (lines 202-205)? Authors may need to discuss clearly reasons for the differences in ichthyoplankton assemblages between spring and autumn.
As shown the above, the manuscript includes some ambiguities and needs to be revised further.

Reviewer 2 ·

Basic reporting

This paper examined the autumn ichtyoplankton assemblages and impacts of the environmental factors. It provides basic information for understanding ichtyoplankton assemblages pattern in Yangtze Estuary at autumn. This paper meets the journal standard.

Experimental design

The analysis was based on seven years surveys lasted twelve years from 1998-2009, but the authors do not show the detail data information for each year. To judge the suitability of the analysis, it seems necessary and important to show the yearly survey information such as number of survey stations, total number of ichtyoplankton, CPUE etc, Because this study used seven years data which lasted for a decadal long, the analysis results may be changed largely depending on the large yearly variations on data, particularly the assemblages will be changed in inter-annual scale.

Validity of the findings

There are no discussions to using references from Yangzi Estuary to supporting this findings, although much discussions are general.

Additional comments

1) Zhang et al . (2015) indicated three assemblages in spring, that are inner, central, and shelf assemblages. In this paper, the autumn ichthyoplankton assemblages are estuary, coastal, and offshore assemblages. What is the difference between the spring and autumn?

2) The title of the paper should be changed as the following in the context of Zhang et al (2015).
Autumn ichthyoplankton assemblage in the Yangtze Estuary shaped by environmental factors .

---

## Round 0.2 · Minor Revisions

The paper needs still some modifications before been accepted. Please, consider the provided suggestions in the revised manuscript.

Reviewer 1 ·

Basic reporting

No comments

Experimental design

No comments

Validity of the findings

No comments

Additional comments

The title and aims have been clarified further in the revised manuscript. Although methods for measuring environmental conditions remain unclear for non-Chinese natives because the supplemental file on the measurement is written by Chinese, I recognized that all measurements has been done based on the manner established by China. Discussion on seasonal differences in ichthyoplankton communities improves the quality of this manuscript. The manuscript would be ready for publication with minor changes described below.

Line 60. Delete one of ‘Yangtze’.
Line 80. It would be better to replace Mo/Yr shown in Line 80 to after ‘during seven autumns’ in Line 77.
Line 195. The reference should be Hernandez-Miranda et al. (2003).
Line 239. Abiotic and biotic.
Line 263. No ‘Wang & Huang 2002’ is found in the reference. It may be Wan et al. (2002) as shown in Line 265. Please make sure.
Line 282. Delete one of ‘in the’.
Lines 333-335. A reference ‘Islam et al. (2006)’ is duplicated.
Lines 339-341. A reference ‘Jiang et al. (2006)b’ is almost same as ‘Jiang et al. (2006)a’. Please make sure. I was not able to find this reference on the journal ‘Marine Environmental Science’.

---

## Round 0.3 · accepted · Accept

Thank you for improving your manuscript.